# The Promising Role of Synthetic Flavors in Advancing Fish Feeding Strategies: A Focus on Adult Female Zebrafish (*Danio rerio*) Growth, Welfare, Appetite, and Reproductive Performances

**DOI:** 10.3390/ani14172588

**Published:** 2024-09-05

**Authors:** Federico Conti, Ike Olivotto, Nico Cattaneo, Massimiliano Pavanello, İdris Şener, Matteo Antonucci, Giulia Chemello, Giorgia Gioacchini, Matteo Zarantoniello

**Affiliations:** 1Department of Life and Environmental Sciences, Università Politecnica delle Marche, 60131 Ancona, Italy; f.conti@pm.univpm.it (F.C.); n.cattaneo@pm.univpm.it (N.C.); max97pavanello@gmail.com (M.P.); idris_943@hotmail.com (İ.Ş.); g.chemello@univpm.it (G.C.); giorgia.gioacchini@univpm.it (G.G.); 2Independent Researcher, Via Pola 18, 64014 Martinsicuro, Teramo, Italy; matteo.antonucci.89@outlook.it

**Keywords:** feed attractant, aquaculture, histology, feed intake, sustainable aquafeeds

## Abstract

**Simple Summary:**

Natural feed attractants are regularly used in aquafeeds to improve their palatability but show a number of disadvantages related to their natural origin, such as inconsistent quality of the raw materials, variable availability, and their dependency from natural sources. Contrariwise, synthetic flavors obtained through standardized processes are emerging as novel, more sustainable and effective substitutes to natural ones. The aim of the present study was to evaluate the use of synthetic flavors as novel feed attractants in zebrafish (*Danio rerio*) diet. A multidisciplinary approach was used to evaluate the dietary effects of the tested flavors—two attractive and one repulsive—on fish feed intake and growth, as well as on appetite regulation, brain reward system, and reproductive performances. Both diets containing the attractive flavors enhanced zebrafish feed intake and growth, promoting the fish reproductive performances without affecting fish welfare. Having ingested more feed, fish were able to invest to a greater extent in high-metabolic-demanding processes such as growth and reproduction. Improving the feeding practices of aquaculture, especially obtaining a quick feeding response by the fish, is a critical aspect for aquafeed production to minimize both the economic and environmental impacts.

**Abstract:**

The present study aimed to test over a six-month period different synthetic flavors in zebrafish (*Danio rerio*) as an experimental model. Specifically, two attractive and one repulsive synthetic flavors were added (1% *w*/*w*) to a specific zebrafish diet, which was administered to the fish during the whole life cycle (from larvae to adults), to evaluate their physiological responses, emphasizing fish welfare, feed intake, growth, reward mechanisms, and reproductive performances. Fish welfare was not affected by all tested flavors, while both attractive flavors promoted fish feed ingestion and growth. The results were supported by both molecular and immunohistochemical analyses on appetite-regulating neurohormonal signals, along with the influence of the feed hedonic properties induced by the brain reward sensation, as demonstrated by the dopamine receptor gene expression. Finally, the present study demonstrated that a higher feed intake also had positive implications on fish reproductive performances, suggesting a promising role of synthetic flavors for the aquaculture industry. In conclusion, the results highlighted the potential of synthetic flavors to improve fish feeding strategies by providing a consistent and effective alternative to traditional stimulants, thereby reducing dependence on natural sources.

## 1. Introduction

In recent years, promising progress in validating the use of alternative and cheaper aquafeed ingredients, especially with respect to fishmeal, has been made. However, aquafeeds still represent a significant portion of production costs in aquaculture. Novel ingredients encompass plant-derived meals, processed animal proteins obtained by animal by-products and insects, and microbial biomasses [1,2,3,4,5,6,7]. Despite their promising features, they can present poor palatability with consequent negative implications for fish production and farm economics [8,9,10]. Therefore, reducing feed losses through improved feeding practices is becoming crucial [11,12]. In this regard, additive ingredients typically derived from natural sources are regularly included in aquafeeds to improve palatability [13,14]. These ingredients, listed as feed attractants, are crucial in fish diet formulation to enhance feed efficiency [15], as by including feeding stimulants in the diet, fish are more likely to seek and consume the feed faster. In fact, these attractants stimulate the fish’s sensory organs, as taste and olfaction, triggering feeding behavior through the release of chemical stimuli in the water [16]. A continuous exposure to different odors in a mutable aquatic environment requires a highly dynamic olfactory system in teleost that exhibits the widespread plasticity of the olfactory sensory neurons (OSNs) to continuously adapt the response to odorants [17]. As a result, feed attractants can improve feed intake, ultimately contributing to fish growth and overall performance in aquaculture operations.

An effective application of feed attractants should simultaneously have a positive impact on fish feed intake and on parameters related to its regulation, such as peripheral and hypothalamic appetite-regulating hormones [18]. In teleost, appetite control and energy balance, at the central hypothalamus level, are integrated by peripheral endocrine signals via a complex interaction, including orexigenic (neuropeptide Y, ghrelin) [19,20] and anorexigenic (leptin) [18,21,22] hormones. In addition, the expression of neurohormonal signals in the fish’s olfactory system are known to influence the olfactory sensitivity via a nutritional state-dependent mechanism, reflecting the prevailing energy status in the brain [23,24].

Along with the homeostatic regulation, hedonic and motivational neurocircuits are activated during feeding, involving monoaminergic systems triggered by the fish taste system [25,26]. The mesolimbic dopamine system plays a crucial role in feeding behavior by being closely linked to the reward sensation experienced during the consumption of palatable feed, promoting the desire to eat [18,26].

The currently employed feed attractants are mostly represented by marine extracts (e.g., meal derived from mollusks and crustaceans), further posing unsustainability issues [27]. In addition, the attractiveness of these substances depends on their natural origin, resulting in the inconsistent quality of the raw materials and their different availability, freshness, and diverse manufacturing practices applied [28,29]. Differently, specific compounds like L-amino acids, small chain peptides, betaine, and nucleotides are potent attractants for fish and are currently used as attractive alternative substances to the traditional feed attractants [30,31]. However, their application presents limitations and disadvantages since their concentration and interaction with other diet’s components can affect the overall performance of the feed [32]. Moreover, the stimulatory effect varies depending on the fish species and life cycle stage considered, while the optimal dosage for the different species is still poorly investigated in the available scientific literature [31,32].

Synthetic flavors are emerging as novel and effective alternatives to traditional feed attractants. Their synthetic origin and standardized production processes can solve the drawbacks related to the use of marine extracts as feed attractants in terms of (i) sustainability, reducing the utilization of marine resources for aquafeed production, and (ii) attractiveness, reducing the variability related to the inconsistent origin and quality of the raw materials. Particularly, Conti et al. [33] recently demonstrated the efficiency of a number of synthetic flavors, produced through standardized processes, as feed attractants in promoting appetite stimulus, feed ingestion, and growth during zebrafish (*Danio rerio*) early development (from larvae to juveniles).

Monitoring the long-term effects of synthetic feed attractants is crucial for the aquaculture sector, especially considering that fish production involves various life cycle stages. Zebrafish represents a widely accepted model organism for preliminary nutrition studies useful to the aquaculture sector [34,35], and many novel aquafeed ingredients have primarily been tested on this experimental model [36,37,38,39]. Additionally, the well-established knowledge on zebrafish oocyte development and maturation and the relatively short time to reach sexual maturity make this species ideal for monitoring the potential effects of synthetic flavors on its reproductive performances as well [40,41]. In fact, since the reproductive system is well conserved across all fish species [42], zebrafish is a useful model for a comprehensive understanding of the mechanisms involved in broodstock reproduction [34,35].

Furthermore, the reproductive phase is one of the most important stages of fish production, since breeders are responsible for offspring production. In this context, female reproduction is of great interest since it is particularly energy demanding to sustain oocyte maturation through the proper allocation of macromolecules in developing oocytes [43]. The uptake of vitellogenin from maternal liver and its conversion into yolk protein is the major mechanism for oocyte development [44]. Vitellogenin gene expression is therefore a reliable molecular marker for assessing fish reproduction. In particular, the zebrafish *vtg* gene repertoire is characterized by five (*vtg1*, *vtg4*, *vtg5*, *vtg6*, and *vtg7*) type I, two (*vtg2* and *vtg8*) type II, and one (*vtg3*) type III vtg genes [45]. Notably, among type I vitellogenins, *vtg1* is the predominant form, while *vtg2* is the most abundant among type II Vtgs, according to Yilmaz et al. [45]. The vitellogenin allocation in oocytes establishes the transition from oocytes at the previtellogenic stage (defined as I and II) to ones at the vitellogenic stage (defined as III onward) [40,41].

Oogenesis can be affected by several stressors including food availability, ingestion, nutritional composition, and assimilation that should be studied in a deeper way to assess the suitability of a diet to correctly sustain fish reproduction [46,47,48]. 

In this context, the present study aimed to test the effects of different synthetic flavors during a six-month feeding trial on zebrafish (from larvae to adults) to assess their potential role as feed attractants, emphasizing fish feed intake, welfare (defined as the aptitude of an animal to live a natural life and show its natural behavior while maintaining good health in its environment), feed reward mechanisms, and reproductive performances.

## 2. Materials and Methods

### 2.1. Ethics

The experimental procedures were approved (n° 2-1/12/22) by the Ethics Committee of the Marche Polytechnic University (Ancona, Italy) and the Italian Ministry of Health (Aut N° 453/2023-PR), and conducted in accordance with the Italian legislation on experimental animals. An anesthetic (MS222, 1 g/L for fish euthanasia; Merck KGaA, Darmstad, Germany) was used to minimize the suffering of the animals.

### 2.2. Synthetic Flavors and Production of Experimental Diets

The three synthetic flavors used for the present study were produced (To Be Pharma S.r.l., S. Egidio alla Vibrata, Teramo, Italy) and selected according to Conti et al. [33]. Briefly, each synthetic flavor consisted of functional solvent, 1,2-propanediol (propylene glycol; PG) in which different flavoring chemicals were dissolved. Both PG and flavoring molecules have been widely considered as safe for use by both the Flavor and Extracts Manufacturers Association and the Food and Drug Administration (Agency for Toxic Substances and Disease Registry 2007).

In this study, two attractive flavors, namely F25 (cheese flavor) and F35 (caramel flavor), and one repulsive flavor, named F32- (coconut flavor), were produced in compliance with the current sector-specific legislation, specifically Regulation (EC) No 1334/2008 and 1333/2008. Additionally, the technical data sheet and the Material Safety Data Sheet (MSDS) confirm the safety of these flavors for both humans and animals. This information ensures that all the flavors used in the study meet regulatory standards and are considered generally safe.

As a control diet (CTRL), a commercial zebrafish diet (Zebrafeed; Sparos LDA, Olhão, Portugal) was selected. Then, four experimental diets were prepared starting from the CTRL diet as follows: (i) PG diet, obtained by adding PG to the CTRL diet; (ii) F25 diet, obtained by adding flavor to the CTRL diet; (iii) F35 diet, obtained by adding F35 flavor to the CTRL diet; (iv) F32- diet, obtained by adding F32- flavor to the CTRL diet. All the synthetic flavors and PG were added to the CTRL diet at a concentration of 1% (*w*/*w*).

### 2.3. Experimental Design

Zebrafish embryos were obtained from the University AB strain broodstock, collected, and maintained (Tecniplast system; Varese, Italy) at pH 7 ± 0.1, 28 ± 0.5 °C, concentrations of ammonia and nitrite < 0.01 mg/L, nitrate concentration < 10 mg/L, and 12L/12Dof photoperiod. After embryo development (48 h), 9000 alive embryos were collected (selected under a steromicroscope) and randomly divided into the six experimental groups (3 tanks per experimental group with 500 embryos/tank) and fed according to Conti et al. [33]: (i) CTRL group; (ii) PG group, zebrafish fed PG diet; (iii) F25 group, zebrafish fed F25 diet; (iv) F35 group, zebrafish fed F35 diet; (v) F32- group, zebrafish fed F32- diet (negative control); (vi) rotation (ROT) group, zebrafish fed F25 and F35 diets, each one singularly administered in a weekly rotation scheme. The inclusion of the ROT group was intended to avoid the possible adaptation of fish olfactory receptors in response to repeated stimulation, as reported by Zufall and Leinders-Zufall [49]. At the beginning of the experiment, 18 tanks (20 L, 3 tanks per experimental group with 500 fish per tank, 1500 per dietary group) with the same broodstock’s water conditions were used to rear zebrafish during the larval stage, with filtration and light set up as described in Olivotto et al. [50]. After 30 days post fertilization (dpf), fish from each tank were gently moved to other tanks (100 L, 3 tanks per experimental group) provided with mechanical and biological filtration (Panaque, Capranica, Italy).

From 5 dpf to the end of the trial (180 dpf), zebrafish were fed the experimental diets(3% body weight, divided in two daily feedings, according to Rashidian et al. [51]).

During the feeding trial, feed particle sizes were <100 μm from 5 to 15 dpf, 101–200 μm from 16 to 30 dpf, 201–400 μm from 31 to 60 dpf, and 401–600 μm from 61 to 180 dpf. Additionally, *Brachionus plicatilis* rotifers (5 individuals/mL) were provided to zebrafish larvae of all the experimental groups from 5 to 10 dpf twice a day, according to Zarantoniello et al. [52]. During the trial, tanks were cleaned daily, and dead specimens, if present, were collected and counted.

At the end of the trial, 144 zebrafish females were sampled, after a lethal dose of MS222 (1 g/L, Merck KGaA), and whole head, brain, liver, intestine, and ovary were sampled and properly stored for the laboratory analysis. Particularly, (i) liver, intestine, and ovary from 5 female zebrafish per tank were fixed in Bouin’s solution for histological analysis; (ii) whole head from 5 female zebrafish per tank were fixed in formalin for himmunohistochemistry; (iii) liver, intestine, and brain from 3 female zebrafish per tank were stored at –80 °C for molecular analysis. Further information is detailed in the dedicated paragraphs.

### 2.4. Biometry and Reproductive Parameters

At the end of the trial (6 months), 20 fish from each tank (60 per experimental group) were weighted with a OHAUS Explorer analytical balance (Greifensee, Switzerland) with an accuracy of 0.1 mg.

Additionally, ovaries were individually weighted to calculate gonadosomatic (GSI) index applying the following formula.
GSI = [(ovary weight/whole body weight) × 100](1)

Survival rate was calculated (at 6 months) by subtracting the dead fish (daily calculated over the whole experimental period) to the initial number of fish per tank.

### 2.5. Feed Intake

The amount of feed ingested by six-month-old zebrafish was evaluated during a 10-day period, according to Conti et al. [33]. During the experiment, a pre-weighed quantity of feed was administered, corresponding to 3% of their body weight, to each tank belonging to the different experimental groups. The eventual uneaten feed in each tank was recovered 15 min post administration of the experimental diets by siphoning, and then oven-dried overnight at 40 °C for quantification. The 15 min duration was chosen in accordance with Conti et al. [33], along with the fact that an effective feed attractant must stimulate a rapid ingestion of the feed within the first minutes of administration [53].

### 2.6. Egg Collection and Hatching Rate

For a 10-day period, six-month-old zebrafish (7 males and 5 females from each experimental tank, in triplicate) were randomly collected daily and transferred to breeding tanks (Tecniplast System; 9 breeding tanks per experimental group), in accordance with Randazzo et al. [54]. During both the acclimatizing (which lasted one week) and breeding periods, fish were fed experimental diets and maintained under the same conditions of the experimental tanks. From each breeding tank (30 min after the onset of light), eggs were collected daily and counted under a stereomicroscope (Leica Wild M3B; Leica Microsystems, Wetzlar, Germany) for the evaluation of daily spawned eggs. Subsequently, only fertilized eggs that had a well-developed blastodisc at 3 h post fertilization were properly stored and used for the hatching test. One hundred and fifty embryos from each experimental group (in triplicate) were randomly selected and transferred to Petri dishes (50 embryos per dish) with E3 medium, prepared as described in Randazzo et al. [54]. Embryos were kept in an incubator at 28 °C for 5 dpf for evaluation. Hatching rate was calculated as hatched eggs/total fertilized eggs × 100%. The hatched larvae were finally counted (number of alive fish at 5 days) and the hatching rate was calculated.

### 2.7. Histological Analysis

Livers, intestines, and ovaries from 5 female specimens per tank (15 per experimental group) were processed using the procedures described in Randazzo et al. [55].

Three transversal sections per fish (15 intestines per experimental group), collected at 50 μm intervals, were analyzed to evaluate potential alterations in intestinal tract and hepatic parenchyma. Specifically, regarding intestines, analyses were conducted on all the undamaged and non-oblique folds and both measurements and histopathological indexes were applied in accordance with Zarantoniello et al. [52] and using ZEN 2.3 software (Zeiss, Oberkochen, Germany). Index scores were assigned as follows: (i) inflammatory influx: + = scarce lymphocytes infiltration, ++ = moderate lymphocytes infiltration, +++ = diffused lymphocytes infiltration; (ii) mucosal folds fusion: + = 0–3 observations per section, ++ = 3–10 observations per section, +++ = >10 observations per section.

Regarding the liver, the fat fraction percentage was quantified on areas with no blood vessels or bile ducts using ImageJ software (ver. 1.54d), setting a homogeneous threshold value, and expressed as percentage of the area occupied by fat on the total hepatic parenchyma (images acquired at 400×).

From each fully sectioned ovary (15 per dietary group), a significant number of histological sections was collected at 300 μm intervals. Subsequently, on stained sections, previtellogenic (classes I and II), class III, class IV, and class V oocytes were counted, according to their morphological features, as described in Randazzo et al. [54]. Oocyte count was performed by using a Zeiss Axio Imager.A2 (Oberkochen, Germany) microscope and ZEN 2.3 lite Software (Zeiss), marking oocytes in each section to minimize repetition, in accordance with Chemello et al. [2].

### 2.8. Immunohistochemistry Analysis

Whole heads from 5 adult zebrafish per tank (in triplicate, 15 per experimental group) were collected at the end of the experiment. Heads were fixed in formalin solution (Bio Optica). The paraffin embedding step was performed according to Randazzo et al. [55]. Sections of 5 µm in thickness were cut from the solidified paraffin blocks using a microtome (Leica RM RTS). The sensory area of zebrafish olfactory epithelium was identified as described in Villamayor et al. [56]. In particular, the typical olfactory rosette was identified at its maximum expression when divided into two symmetrical zones by the presence of a wide central stalk that serves to support the lamellae. Lamellae, located in the central and medial region of the rosette, are composed of a continuous sensory area, as well as a lateral non-sensory epithelium. The sensory area comprises a characteristic pseudostratified columnar epithelium formed primarily by olfactory sensory neurons (OSNs), as well as basal and supporting cells [57,58,59].

According to Lombó et al. [60], slices were incubated in a buffer containing 10 mM sodium citrate and 0.05% (*v*/*v*) Tween at 100 °C for 20 min to perform an antigen retrieval. The sections were then permeabilized in 1% PBS-T (triton X-100) for 20 min after washing them with distilled water. One hour blocking was carried out in PBS supplemented with 3% BSA and 0.1% (*v*/*v*) Tween. Polyclonal anti-NPY antibody (1:5000; Merck KGaA, Cat. No. N9528, 2018) was used as primary antibody for sections incubation, performed overnight at 4 °C. Subsequently, sections were incubated with a Goat Anti-Rabbit IgG H&L (Alexa Fluor^®^ 488; ab150077, Abcam, Cambridge, UK) at 37 °C for 1 h after three washing steps with PBS. In the final step, slides were mounted with DAPI-Aqueous Fluoroshield (ab104139, Abcam). The images were acquired in five different sections per sample using a confocal microscope Nikon A1R, collected at 20 μm intervals. For the qualitative analysis of the NPY-expressing neurons, found in the sensory area of the lamellae, scores were assigned as follows: + = scarce NPY-expressing cells distribution, ++ = moderate NPY-expressing cells distribution, +++ = diffused NPY-expressing cells distribution.

The specificity of the anti-NPY antibody (Sigma-Aldrich, St. Louis, MI, USA) used in this study was previously demonstrated by Kaniganti et al. [61].

### 2.9. Molecular Analyses

The gene expression of target genes in brain, liver, and whole intestine samples from 3 female fish per tank (in triplicate, 9 brains, 9 livers, and 9 intestines per group) was measured through real-time PCR. Total RNA extraction and purification and cDNA synthesis were performed according to Olivotto et al. [62] and following the manufacturers’ protocols. The integrity of RNA was assessed by running 1 µg of total RNA stained with GelRedTM on a 1% agarose gel.

Real-time quantitative PCR (qPCR) reactions were performed in an iQ5 iCycler thermal cycler (Bio-Rad, Hercules, CA, USA) on a 96-well plate, following the parameters and reagents reported in Cattaneo et al. [63]. Relative mRNA abundance was used to quantify the gene expression using, as housekeeping genes (two), the ribosomal protein L13 (*rpl13*) and actin-related protein 2/3 complex subunit 1a (*arpc1a*) to standardize the results. The relative quantification of genes involved in fish growth (insulin-like growth factor, *igf1*; myostatin, *mstnb*), appetite regulation (ghrelin, *ghrl*; neuropeptide Y, *npy*; and leptin, *lepa*), brain reward system (dopamine receptor D2a, *drd2a*), stress response (glucocorticoid receptor, *nr3c1*), and vitellogenin production (vitellogenin 1, *vtg1*; vitellogenin 2, *vtg2*; vitellogenin 3; *vtg3*) was performed on samples from target organs. Particularly, (i) *igf1*, *mstnb*, *nr3c1*, *lepa*, *vtg1*, *vtg2*, and *vtg3* were tested in liver samples; (ii) *ghrl* was tested in intestine samples; and (iii) *npy* and *drd2a* were tested in brain samples. The primer sequences used to evaluate the gene expressions in this study are reported in Table 1. The efficiency was around 90% for all the tested primers, with an R^2^ that ranged from 0.995 to 0.998. The fluorescence was monitored at the end of each cycle, and the melting curve analyses revealed one single peak for every product. Two no-template controls (NTCs) were added in each run to guarantee the absence of contamination in the Master Mix. Amplification products were sequenced, and homology was verified. The mRNA levels of target genes were calculated using the geometric mean of two stably expressed reference genes using the Bio-Rad CFX Manager 3.1 software. Gene expression changes among groups are reported as relative mRNA abundance. The qPCR data were processed using iQ5 optical system software version 2.0 (Bio-Rad), including GeneEx Macro iQ5 Conversion and genex Macro iQ5 files.

### 2.10. Statistical Analyses

Normality (Shapiro–Wilk test) and homoscedasticity (Levene’s test) were checked for all data. ANOVA followed by Tukey’s multiple comparison post hoc test (software package Prism 8; GraphPad software version 8.0.2, San Diego, CA, USA) was applied to analyze all the data from the different analyses (significance set at *p* < 0.05).

## 3. Results

### 3.1. Biometry

Regarding survival rate, no significant differences were evident among the experimental groups (87 ± 4; 85 ± 5, 86 ± 4, 85 ± 3, 86 ± 4, and 84 ± 3% for CTRL, PG, F25, F35, ROT, and F32-, respectively). Figure 1 shows the final body weight (FBW) of specimens fed the experimental diets. CTRL and PG groups did not show significant differences. These groups instead showed a significantly (*p* < 0.05) lower FBW compared to the other groups. In addition, specimens from the F35 group showed a significantly (*p* < 0.05) higher FBW compared to all the other experimental groups. Finally, the F25 group showed a significantly (*p* < 0.05) higher FBW compared to the ROT group.

### 3.2. Feed Intake

Figure 2 shows the percentages of feed ingested by adult zebrafish fed the experimental diets. No significant differences were detected between the CTRL and PG groups, while both the F25 and F35 groups showed a significantly (*p* < 0.05) higher feed intake compared to the CTRL, PG, and F32- groups. No significant differences were evident among the remaining groups.

### 3.3. Liver and Intestine Histomorphology

Regarding the intestine and liver of specimens from the different experimental groups, no remarkable alterations in the tissues’ morphology or any signs of inflammation were observed (Figure 3). Corroborating this, no significant differences were detected in all histopathological indexes analyzed in the intestine (Table 2). Additionally, physiological fat accumulation was observed in the hepatic parenchyma of all experimental groups, further supported by the statistical quantification of the fat fraction percentage calculated on liver sections, which showed no significant differences among groups (41.5 ± 6.1, 36.9 ± 1.8, 32.9 ± 10.1, 22.9 ± 6.1, 39.2 ± 2.9, 22.1 ± 3.2%, for CTRL, PG, F25, F35, ROT, and F32-, respectively; *p* > 0.05).

### 3.4. Real-Time PCR

*Growth.* As regards *igf1*, no significant differences were detected between the CTRL and PG groups (Figure 4a). Conversely, the F35 and ROT groups showed a significantly (*p* < 0.05) higher expression of *igf1* compared to the CTRL, PG, and F32- groups (no significant differences among them).

Regarding *mstnb*, no significant differences were detected among the different experimental groups (Figure 4b).

*Appetite.* Considering *npy* gene expression (Figure 5a), the CTRL and PG groups did not show significant differences among them. Conversely, the F25 and F35 groups were characterized by a significantly (*p* < 0.05) lower *npy* expression compared to the other groups.

With respect to *ghrl* gene expression (Figure 5b), fish fed the CTRL, PG, and F35 diets did not show significant differences among them, while CTRL and F35 showed a significantly (*p* < 0.05) higher *ghrl* expression compared to the other groups. Additionally, the F25 group showed a significantly (*p* < 0.05) lower *ghrl* expression compared to all the other experimental groups.

Finally, considering *lepa* gene expression (Figure 5c), the CTRL, PG, F25, and F35 groups did not evidence significant differences among them, while showing a significantly (*p* < 0.05) higher expression compared to the ROT and F32- groups.

*Brain dopaminergic activity.* The CTRL and PG groups showed no significant differences among them in *drd2a* expression (Figure 6). Conversely, the F25 and F35 groups showed significantly (*p* < 0.05) lower expression of *drd2a* compared to all other groups, excluding the ROT one.

*Stress response.* No significant differences were observed among experimental groups in the *nr3c1* gene expression (Figure 7).

### 3.5. Immunohistochemistry in the Olfactory Epithelium

The qualitative analysis performed on the zebrafish olfactory epithelium (OE) revealed NPY-expressing olfactory neurons (OSNs) in all the different experimental groups. However, the CTRL, PG, ROT, and F32- groups showed a homogeneous and diffused distribution of NPY-expressing neurons along the sensory epithelium of the lamellae (Figure 8d and Table 3). Conversely, both the F25 and F35 groups showed a scarce and heterogeneous distribution of NPY-expressing neurons in the sensory area along the lamellae (Figure 8e and Table 3).

### 3.6. Reproductive Performances

#### 3.6.1. Gonadosomatic Index

Considering the gonadosomatic index (Table 4), no significant differences were detected among the experimental groups (*p* > 0.05).

#### 3.6.2. Histological Analysis of Ovaries

Histology performed on ovary sections of all experimental groups revealed the presence of different oocyte classes (Figure 9). In particular, for the oocyte count, according to Randazzo et al. [54], oocytes were assigned to a pre-vitellogenic (PreV; stages I and II) or post-vitellogenic (PostV; stage III onward) stage based on their morphological features: previtellogenic oocytes, with or without the cortical alveus presence (100–280 µm diameter); class III oocytes, characterized by enlarged yolk vesicles and vitellin membrane (280–740 µm diameter); and class IV oocytes, characterized by nuclear envelope break down and opaque ooplasm (>740 µm diameter). Since the class V oocytes ovulation occurs in few hours, they were rarely found in the analyzed zebrafish ovaries [40,41].

As reported in Table 5, both the F25 and F35 groups showed a significantly (*p* < 0.05) lower percentage of pre-vitellogenic oocytes (72.19 ± 2.90 and 75.37 ± 2.69%, respectively) and a significantly (*p* < 0.05) higher percentage of post-vitellogenic oocytes (27.08 ± 2.89 and 24.63 ± 2.69%, respectively) compared to all other groups, which instead showed no significant differences among them (*p* > 0.05).

#### 3.6.3. Spawned Eggs and Hatching Rate

As shown in Figure 10a, the F25 and F35 groups were characterized by a significantly higher number of daily spawned eggs compared to the CTRL, PG, ROT, and F32- groups within the 10-day period (485 ± 50, 485 ± 77, 576 ± 73, 579 ± 58, 388 ± 32, and 450 ± 82, for CTRL, PG, F25, F35, ROT, and F32-, respectively). Additionally, the ROT group was characterized by a significantly lower number of daily spawned eggs compared to the CTRL and PG groups.

Regarding the hatching rate (Figure 10b), no significant differences were observed among the experimental groups (84.1 ± 6.1, 87.9 ± 5.5, 84.5 ± 9.3, 86.3 ± 14.5, 79.8 ± 12.2, 80.7 ± 11.3%, for CTRL, PG, F25, F35, ROT, and F32-, respectively). 

Finally, no significant differences were detected in the percentage of unfertilized eggs, which was approximately 15% for all groups, in accordance with previous studies [64].

#### 3.6.4. Vitellogenin Gene Expression

Considering *vtg1* (Figure 11a), no significant differences were evident between the CTRL and PG groups, while F25 was characterized by a significantly (*p* < 0.05) higher *vtg1* expression compared to all the other groups, except for F35.

Considering *vtg2* (Figure 11b), no significant differences were detected among the CTRL and PG groups. The F25 and F35 groups showed a significantly (*p* < 0.05) higher *vtg2* expression compared to both the CTRL and F32- groups. 

Finally, considering *vtg3* (Figure 11c), no significant differences were detected between the CTRL and PG groups, while the F35 group showed a significantly (*p* < 0.05) higher *vtg3* expression compared to all the other groups that did not show significant differences among them.

## 4. Discussion

Improving feeding techniques is a key aspect in fish farming to ensure better production yields that eventually have positive implications for both the sector’s economy and the environment.

Considering this, the present study aimed to assess the role of synthetic flavors as potential and more sustainable feed attractants in a 6-month feeding trial on zebrafish, focusing on the female adult stage physiological responses. In fact, the zebrafish model has been considered suitable for studies for finfish aquaculture research, allowing us to address all aspects of the life cycle, including nutrition, reproduction, and welfare [35].

According to Conti et al. [33], both the solvent (PG) and all the tested synthetic flavors had no adverse effects on the overall fish health and on the target organs usually investigated when novel ingredients/additives are included in fish diets. In fact, no alterations in the intestinal architecture or differences in the histopathological indexes examined were detected; a physiological structure of the hepatic parenchyma was observed in all the experimental groups, and no significant differences in the expression of the stress-related marker (*nr3c1*) was evidenced.

Apart from suggesting that the application of the present synthetic flavors did not affect fish welfare, the flavors’ attractive effect was confirmed during the feed intake experiment. Particularly, the maximum feed intake achieved was observed in fish fed diets implemented with cheese (F25) and caramel (F35) flavors. Maximizing feed consumption in aquaculture is a remarkable achievement, as rapid and complete feed intake allows for faster fish growth, thus shortening production time while decreasing feed losses [15,53]. These results were further supported by the higher final body weight of the fish and the consequent upregulation of the growth-promoting factor *igf1*, as already demonstrated by Conti et al. [33] during zebrafish early development.

It is well established that fish feeding behavior is driven by appetite regulatory mechanisms as a consequence of the interaction between signals (both orexigenic and anorexigenic) produced at the gastrointestinal tract and brain level [18,20]. In addition, foraging is influenced by a wide range of water-soluble compounds which are discriminated by the fish olfactory system [65]. Neurohormonal signals can influence this peripheral circuit, including the action of the olfactory sensory neurons (OSNs), via a nutritional state-dependent modulation [61]. For example, a reduction in Neuropeptide Y signaling in the fish olfactory system has been associated with higher nutrient availability [23,24], while an increase in the number of NPY-expressing OSNs was associated with starvation [61]. On this basis, the lower brain *npy* gene expression level coupled with the lower NPY immunoreactivity detected in the zebrafish olfactory epithelium sustained the higher satiety state of groups fed both cheese (F25)- and caramel (F35)-flavored diets.

Nutrient availability, as well as malnutrition and starvation, are known to influence one of the most powerful orexigenic signals synthetized at the peripheral level, the ghrelin [19]. Increasing levels of this hormone, synthetized in the intestine, are known to stimulate *npy* expression at the central level [20]. Thus, a higher feed intake can decrease peripheral *ghrl* expression, which in turn lowers *npy* expression at the central level. This *scenario* was evident only in the group fed the cheese-flavored diet, confirming a more satiated state in this group. Considering the group fed the caramel-flavored diet, *npy* expression at the central level supported the above-mentioned outcomes, while a non-obvious result was obtained for *ghrl* gene expression. This latter result can be related to the fact that other physiological mechanisms, such as the gastrointestinal tract motility, digestive, and absorption functions, can have a deep impact on *ghrl* gene expression in the intestine [66,67,68].

Regarding leptin, its gene expression in groups CTRL, PG, F25 (cheese), and F35 (caramel) suggested an adequate nutritional status of the fish, since the levels of this hormone are directly related to the body energy reserves and thus feed intake [21,22].

Appetite in teleost is also strictly related to the monoaminergic system that influences feeding behavior along with the above-mentioned homeostatic neuroendocrine pathways [25]. A highly palatable feed can promote its ingestion because of its hedonic properties, actively involving the dopamine receptors in fish brain reward system [26]. Among the different dopamine receptors, Drd2 is suggested to be strongly related to feed intake in vertebrates [69,70]. Additionally, during a time-prolonged positive stimulation, dopamine receptor downregulation has been associated with the reinforcement mechanisms [71]. Accordingly, in the present study, which lasted for six months, *drd2a* was downregulated in caramel and cheese groups, supporting the attractive effectiveness of the F25 and F35 flavors as positive stimuli.

Feed intake is also closely related to the reproductive functions in teleost. In fact, it is well established that malnourishment conditions or scarce feed/nutrient availability can have negative implications on fish reproduction [47]. The quality of maternal nutrition is crucial, especially during oocyte development, a process that requires the proper allocation of macromolecules such as vitellogenin [46,72]. In this regard, oocyte development is well known in zebrafish. In stage I, oocytes begin to enlarge, and follicles start to form. Subsequently, cortical alveoli accumulate within the oocytes (Stage II), while stage III oocytes undergo vitellogenesis, resulting in a significant dimensional increase. In stage IV oocyte, yolk vesicles begin to fuse with each other, resulting in oocyte maturation. Finally, when mature eggs are ovulated and ready for spawning, this is referred to as stage V [40,41].

Histological analysis on the ovaries from the cheese (F25) and caramel (F35) groups evidenced a high number of post-vitellogenic oocytes, reflecting an increased maturation state. In addition, since the vitellogenin uptake in oocytes establishes their transition from previtellogenic to mature stages [40,41], this latter result was further supported by *vtg* gene expression. In fact, the detected upregulation of all VTGs isoforms analyzed in specimens fed the cheese and caramel diets suggested the promotion of the oocyte maturation process in these experimental groups, which was further supported by a higher spawning rate. Having ingested more feed, specimens from these groups were likely more capable to invest in high-metabolic-demanding physiological mechanisms such as reproduction [43]. This result may also be explained by the fact that such specimens produced more eggs as a consequence of their bigger size [73], reflecting that a higher feed intake supported the growth and reproductive capabilities of the specimens. Nevertheless, despite the differences in the average daily number of spawned eggs among groups, a comparable hatching rate was observed, a result easily explainable by the fact that the administered dietary formula was the same in all experimental groups [47]. Finally, in accordance with previous outcomes [33], and considering the overall results obtained in the present study, the ROT (rotation) group was not a reliable feeding scheme. The repeated positive flavor/stimulus shift (cheese/caramel) did not provide the expected positive results since the ROT group was designed to avoid the possible adaptation of fish olfactory receptors in response to a repeated stimulation [49]. Concurrently and in accord to a previous study [33], the repulsive effect of the coconut flavor (F32-) was reduced by the interaction with the feed odor, or by a possible adaptation of the fish to the flavor.

## 5. Conclusions

Both CTRL and PG diets showed comparable results, while the attractive synthetic flavors tested confirmed their attractive role also in fish exposed over a long-term period. The provision of both diets containing cheese- or caramel-attractive flavors resulted in increased feed ingestion via appetite and olfactory stimuli, all mediated by the influence of the feed hedonic properties and a higher growth rate. Moreover, the inclusion of attractive flavors positively promoted fish reproductive performances, as the fish were able to invest to a greater extent in this high-metabolic-demanding process. Finally, in accordance with a previous study [33], the ROT group was not a reliable feeding scheme, and the repulsive coconut flavor led to controversial results.

In conclusion, the present study demonstrated the promising role of synthetic flavors in advancing fish feeding strategies. Using these new-generation additives will reduce the natural dependence on the conventional feeding stimulants, and they rely on a standardized manufacturing process, ensuring consistent attractive effectiveness within aquafeeds.

## Figures and Tables

**Figure 1 animals-14-02588-f001:**
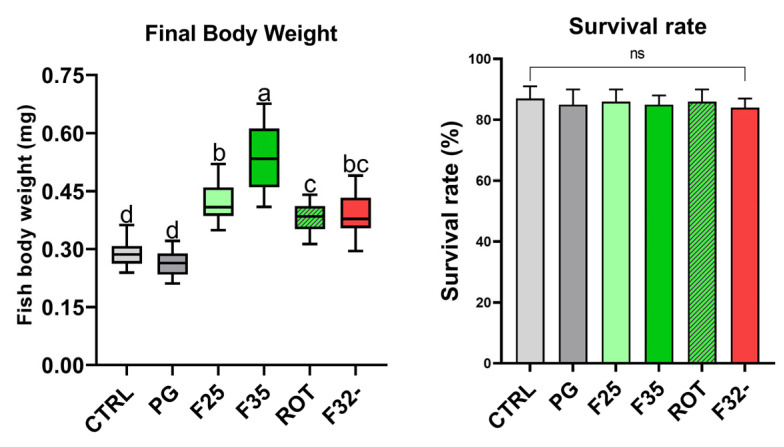
Final body weight (mg) (**left**) and survival rate (%) (**right**) of adult zebrafish fed the experimental diets. Boxplot shows minimum and maximum (whiskers), first quartile, median, and third quartile (box). ^a–d^ Different letters indicate statistically significant differences among the experimental groups (*p* < 0.05). Values are presented as mean ± SD (*n* = 3); ns, no significant differences.

**Figure 2 animals-14-02588-f002:**
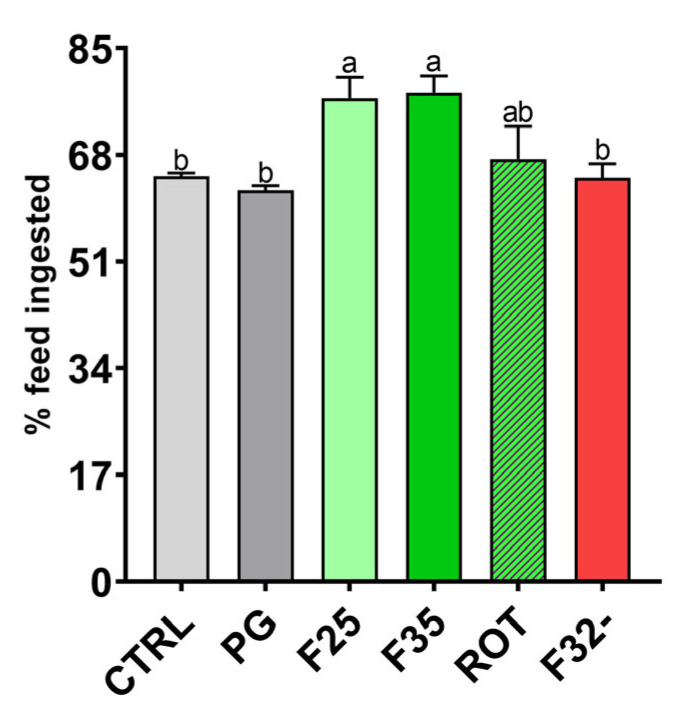
Percentage of feed ingested by adult zebrafish, calculated 15 min post dietary administration. Results are expressed as mean + SD (*n* = 3). ^a,b^ Different letters indicate statistically significant differences among the experimental groups (*p* < 0.05).

**Figure 3 animals-14-02588-f003:**
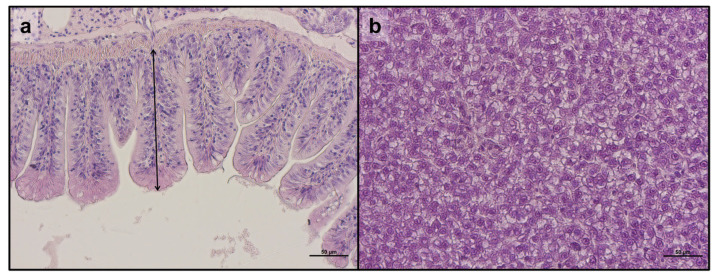
Representative figures of intestine and liver parenchyma of adult zebrafish (**a**,**b**) from the present study: (**a**) details on mucosal folds (double-headed arrow: mucosal folds height); (**b**) liver parenchyma. Scale bars: 50 μm.

**Figure 4 animals-14-02588-f004:**
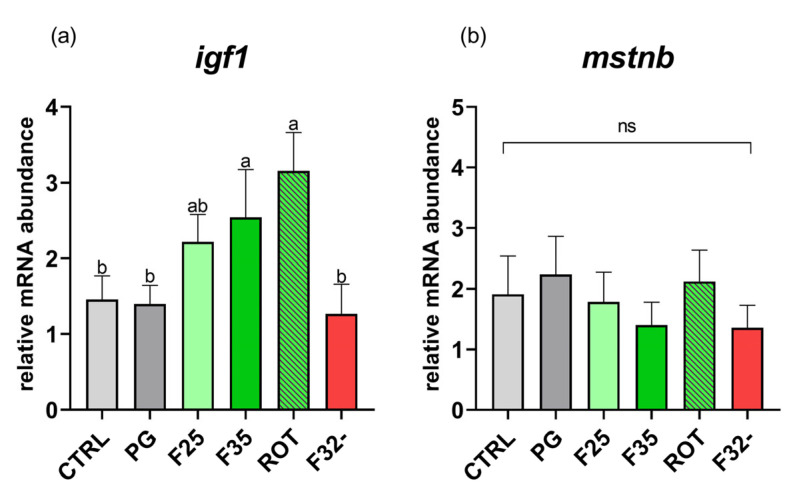
Relative mRNA abundance of genes involved in growth analyzed in liver samples from adult zebrafish. (**a**) *igf1* and (**b**) *mstnb*. Results are expressed as mean + SD (*n* = 5). ^a,b^ Different letters denote statistically significant differences among the experimental groups; ns, no significant differences.

**Figure 5 animals-14-02588-f005:**
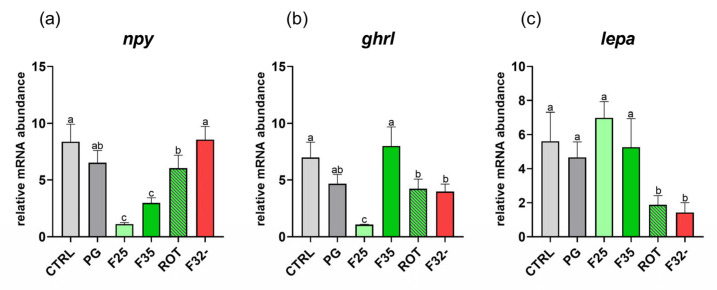
Relative mRNA abundance of genes involved in appetite regulation analyzed in brain (*npy*), intestine (*ghrl*), and liver (*lepa*) samples from adult zebrafish. (**a**) *npy*, (**b**) *ghrl*, and (**c**) *lepa*. Results are expressed as mean + SD (*n* = 5). ^a–c^ Different letters indicate statistically significant differences among the experimental groups.

**Figure 6 animals-14-02588-f006:**
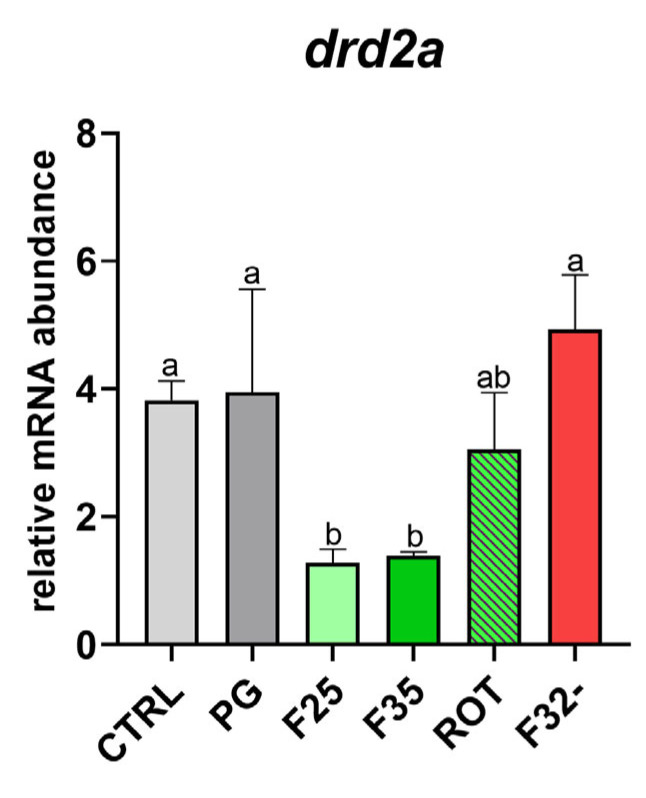
Relative mRNA abundance of *drd2a* involved in brain reward system analyzed in brain samples from adult zebrafish. Results are expressed as mean + SD (*n* = 5). ^a,b^ Different letters denote statistically significant differences among the experimental groups.

**Figure 7 animals-14-02588-f007:**
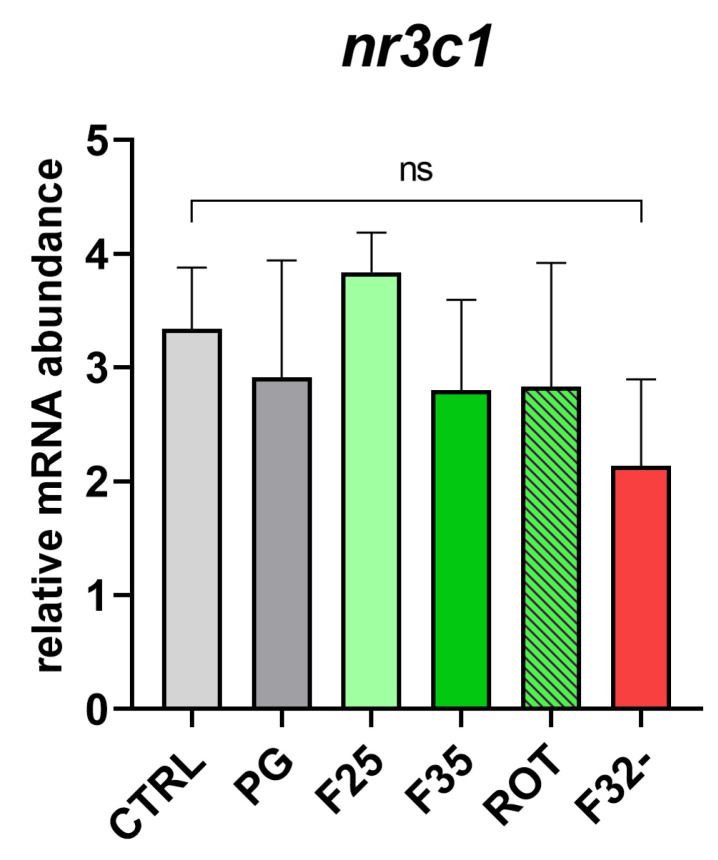
Relative mRNA abundance of *nr3c1* involved in stress response analyzed in liver samples from adult zebrafish. Results are expressed as mean + SD (*n* = 5); ns, no significant differences.

**Figure 8 animals-14-02588-f008:**
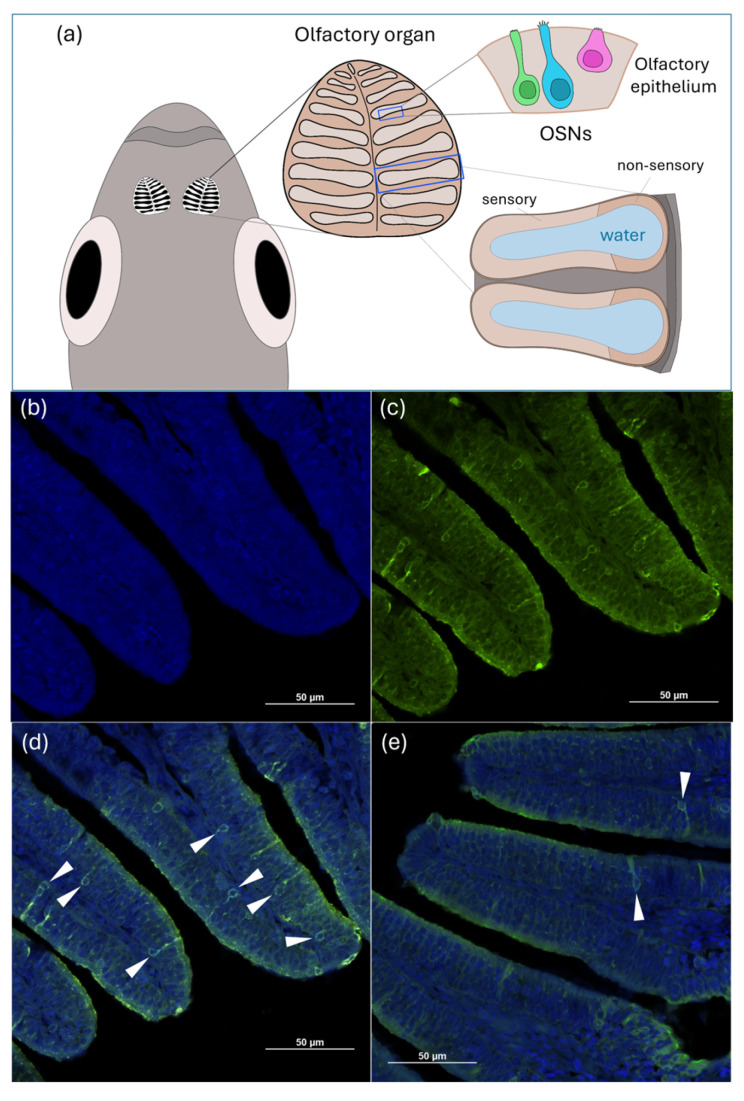
Neuropeptide Y (NPY) expression in the olfactory epithelium of zebrafish fed the different experimental diets. (**a**) Zebrafish olfactory organ, highlighting the morphology of epithelium arranged in lamellae. The central and medial region of each lamella includes a continuous sensory area, in which different olfactory sensory neurons (OSNs) are located, as well as a lateral non-sensory area (illustration by Conti F.); (**b**,**c**) DAPI (blue) correspond to nuclei while green (Alexa Fluor^®^ 488) correspond to NPY (CTRL group); (**d**) representative transverse section of zebrafish olfactory epithelium from CTRL group showing NPY-expressing cells (arrowheads); (**e**) representative transverse section of zebrafish olfactory epithelium from F35 (caramel) group showing NPY-expressing cells (arrowheads). Scale bar = 50 µm.

**Figure 9 animals-14-02588-f009:**
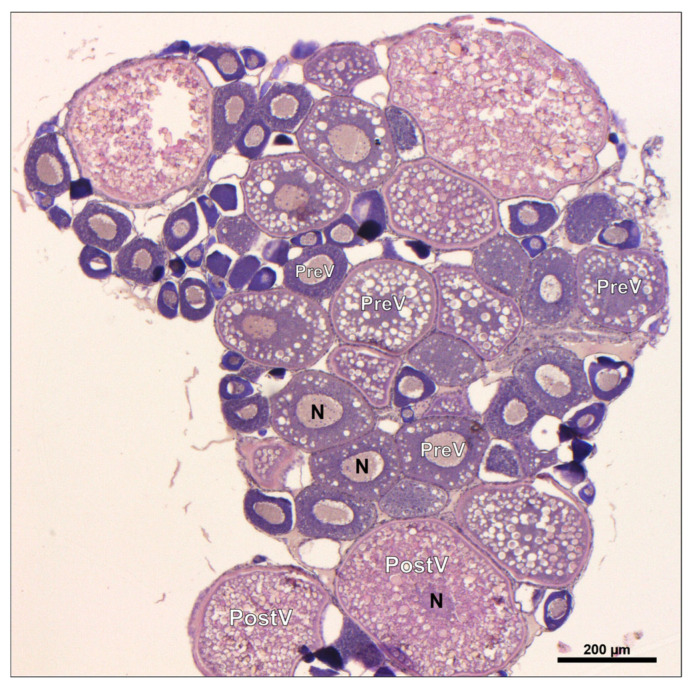
Oocyte developmental stages in zebrafish ovary. PreV, pre-vitellogenic stage; PostV, post-vitellogenic stage; N, nucleus. Scale bar: 200 µm.

**Figure 10 animals-14-02588-f010:**
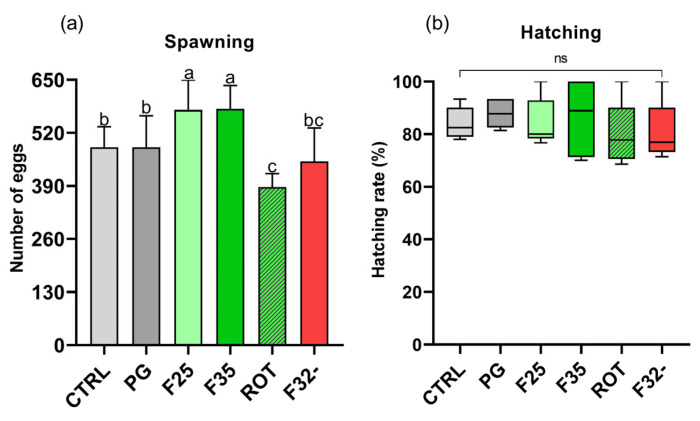
Total number of spawned eggs per day and percentage of hatching rate observed in fish fed the different experimental diets. (**a**) Average daily number of eggs laid within a 10-day period. (**b**) Hatching rate expressed as percentage. Boxplots show minimum and maximum (whiskers), first quartile, median, and third quartile (boxes). ^a–c^ Different letters indicate statistically significant differences between experimental groups (*p* < 0.05). Values are presented as mean ± SD (*n* = 10 spawning days, *n* = 9 for hatching rate).

**Figure 11 animals-14-02588-f011:**
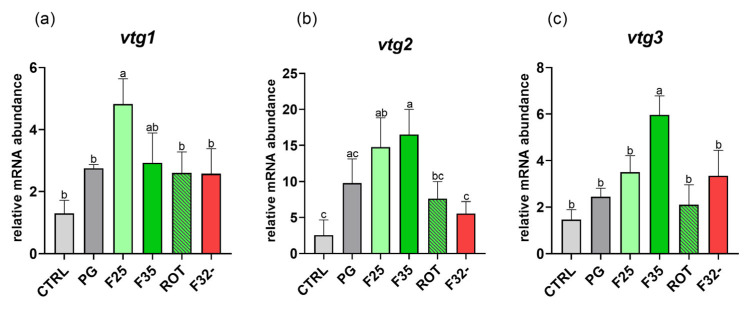
Relative mRNA abundance of genes involved in vitellogenin production analyzed in liver from adult female zebrafish. (**a**) *vtg1*, (**b**) *vtg2*, and (**c**) *vtg3*. Results are expressed as mean + SD (*n* = 5). ^a,b^ Different letters indicate statistically significant differences among the experimental groups.

**Table 1 animals-14-02588-t001:** Primer sequences used in the present study, annealing temperature (AT), and NCBI IDs.

Gene	Forward Primer (5′-3′)	Reverse Primer (5′-3′)	AT (°C)	ID
*igf1*	GGCAAATCTCCACGATCTCTAC	CGGTTTCTCTTGTCTCTCTCAG	53	ZDB-GENE-010607-2
*mstnb*	GGACTGGACTGCGATGAG	GATGGGTGTGGGGATACTTC	58	ZDB-GENE-990415-165
*ghrl*	CAGCATGTTTCTGCTCCTGTG	TCTTCTGCCCACTCTTGGTG	58	ZDB-GENE-070622-2
*npy*	GTCTGCTTGGGGACTCTCAC	CGGGACTCTGTTTCACCAAT	60	ZDB-GENE-980526-438
*lepa*	CTCCAGTGACGAAGGCAACTT	GGGAAGGAGCCGGAAATGT	60	ZDB-GENE-081001-1
*drd2a*	TGGTACTCCGGAAAAGACG	ATCGGGATGGGTGCATTTC	58	ZDB-GENE-021119-2
*vtg1*	GCCAAAAAGCTGGGTAAACA	AGTTCCGTCTGGATTGATGG	59	ZDB-GENE-001201-1
*vtg2*	TGCCGCATGAAACTTGAATCT	GTTCTTACTGGTGCACAGCC	58	ZDB-GENE-001201-2
*vtg3*	GGGAAAGGATTCAAGATGTTCAGA	ATTTGCTGATTTCAACTGGGAGAC	58	ZDB-GENE-991019-2
*nr3c1*	AGACCTTGGTCCCCTTCACT	CGCCTTTAATCATGGGAGAA	58	ZDB-GENE-050522-503
*rpl13 (hk)*	TCTGGAGGACTGTAAGAGGTATGC	AGACGCACAATCTTGAGAGCAG	59	ZDB-GENE-031007-1
*arpc1a (hk)*	CTGAACATCTCGCCCTTCTC	TAGCCGATCTGCAGACACAC	60	ZDB-GENE-040116-1

Abbreviations: *hk*, housekeeping genes.

**Table 2 animals-14-02588-t002:** Histological indexes measured in the intestine of adult zebrafish fed the experimental diets. Data for intestinal mucosal folds height are reported as mean ± SD (*n* = 15).

	CTRL	PG	F25	F35	ROT	F32-
Mucosal folds height	197.9 ± 8.6	213.3 ± 41.4	222.8 ± 54.3	180.2 ± 11.1	177.4 ± 16.1	183.5 ± 20.6
Inflammatory influx	+	+	+	+	+	+
Mucosal folds fusion	+	+	+	+	+	+

**Table 3 animals-14-02588-t003:** Qualitative analysis of the NPY-expressing neurons in the sensory area of the zebrafish lamellae, expressed as mean observation performed in all sections for each group.

	CTRL	PG	F25	F35	ROT	F32-
NPY-expressing cell distribution	+++	+++	+	+	+++	+++

**Table 4 animals-14-02588-t004:** Gonadosomatic indexes of adult female zebrafish fed the experimental diets (GSI).

	CTRL	PG	F25	F35	ROT	F32-
GSI	8.69 ± 2.42	7.86 ± 0.67	7.16 ± 0.88	8.38 ± 1.38	10.16 ± 3.35	12.36 ± 5.02

**Table 5 animals-14-02588-t005:** Percentage of pre-vitellogenic and post-vitellogenic oocytes.

	CTRL	PG	F25	F35	ROT	F32-
PreV	87.73 ± 3.02 ^a^	85.92 ± 3.66 ^a^	72.19 ± 2.90 ^b^	75.37 ± 2.69 ^b^	86.5 ± 1.50 ^a^	87.09 ± 3.95 ^a^
PostV	12.27 ± 3.02 ^b^	14.09 ± 3.66 ^b^	27.08 ± 2.89 ^a^	24.63 ± 2.69 ^a^	12.59 ± 1.13 ^b^	12.31 ± 3.36 ^b^

Results are expressed in percentage ([number of oocytes in each stage/total oocytes] × 100) as mean ± SD. ^a,b^ Different letters denote statistically significant differences among the groups (*p* < 0.05). PreV, pre-vitellogenic stage; PostV, post-vitellogenic stage.

## Data Availability

The data presented in this study are available on request from the corresponding author.

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
