# Peer review of "The Promising Role of Synthetic Flavors in Advancing Fish Feeding Strategies: A Focus on Adult Female Zebrafish (Danio rerio) Growth, Welfare, Appetite, and Reproductive Performances"

_animals, 2024, doi:10.3390/ani14172588_

Round 1

Reviewer 1 Report

Comments and Suggestions for Authors

attached in report

Comments on the Quality of English Language

a through revision is required

Author Response

Many thanks for the comments, but we think that there has been a mistake and the comments included are not referred to our MS.

Because of this we are not able to address them and we hope that by answering the 2 other reviewers comments the paper can be accepted for publication.

Reviewer 2 Report

Comments and Suggestions for Authors

This paper has investigated the systematic flavor effects on fish growth, welfare, appetite and reproduction using zebrafish model. The study has a good novelty, and the story is interesting. The editing, writing and logic are clear to present. Some comments author can improve before publication.

Line 119-123, this paragraph can be deleted since there is no need to introduce more about the detailing of stages. In the discussion, it is enough that authors will do more efficient and clear presentation.

Table1, what is the control house keeping gene for calculating the relative gene expression? Authors should clarify the method for calculate the relative genes expression unit in Line 298-314, if they don’t use the 2 delta CT values methods.

Author Response

Reviewer 2

This paper has investigated the systematic flavor effects on fish growth, welfare, appetite and reproduction using zebrafish model. The study has a good novelty, and the story is interesting. The editing, writing and logic are clear to present. Some comments author can improve before publication.

Many thanks for the positive comments.

Line 119-123, this paragraph can be deleted since there is no need to introduce more about the detailing of stages. In the discussion, it is enough that authors will do more efficient and clear presentation.

As suggested the sentence has been delated from the introduction section. The introduction has been revised accordingly.

Table1, what is the control housekeeping gene for calculating the relative gene expression? Authors should clarify the method for calculate the relative genes expression unit in Line 298-314, if they don’t use the 2 delta CT values methods.

In the Table 1, the two housekeeping genes (rpl13 and arpc1a) were already present indicating that two hkg were used in the present study. Hk was added in the table to further make this aspect evident.

Furthermore, a sentence has been added regarding the method used for gene expression calculation to make this concept clearer: “The mRNA levels of target genes were calculated using the geometric mean of two stably expressed reference genes, using the Bio-Rad CFX Manager 3.1 software. Gene expression changes among groups are reported as relative mRNA abundance. The qPCR data were processed using iQ5 optical system software version 2.0 (Bio-Rad), including GeneEx Macro iQ5 Conversion and genex Macro iQ5 files”.

Reviewer 3 Report

Comments and Suggestions for Authors

L.14: Attractants may be replaced by palatable as it seems more appropriate in the sentence context.

L.17: Explain how the synthetic flavors are sustainable.

L. 19: I think attractants and palatable are two distinct things. Attractants aim to stimulate the fish feed intake while palatable ensure that the fish keep and assimilate the aquafeed. Therefore was the study looking more at attractant or palatability of the synthetic flavors ?

L.20: What was that aim of the repulsive ? Was it like a negative control ?

L.23: Fish welfare should be defined.

L.46: Aquafeed should be define.

L.46-48: This sentence may be rephrased in two sentences to get clearer.

L.48-51: Same comment. Short sentences are clearer. The ideas are well presented but one at the time will make the manuscript better.

L.52: A dash between reference 10 and 11 [10-11], will be more consistent and this applies for the whole manuscript.

L.61: “to adapt continuously the response to odorants”.

L.73: Is it hedonic and motivational ? Hedonic or motivational ?

L.125:  In its ingestion, “its” is confusing. If you decide to use “its”, you need also to ass it in front of quality and assimilation to be more consistent.

L.139: The concentration of MS222 is for euthanasia or anaesthesia ? This should be clarified.

L.149-155: What was the role of the repulsive (F32-) ?

L.192-193: How the different organs were conserved until analysis (snap freeze, RNA later) ? Which organs for which type of analysis ?

L.293: If you describe the DNase treatment, you also need to describe how did you inactive the enzyme post treatment.

L.295: How did you assessed that your RNA was not degraded ?

L.299-301: You need to state whether you just merged the two housekeeping genes or if you selected the most stable of them.

L.309-310: You did not mention the cycling parameters for the qCPR reaction, the kit used, the reaction mix, the type of fluorescent (SYBERGREEN ?), the primers concentration…

L.311: To guarantee absence of contamination from what ? MiliqWater ? Master Mix ?

L.315: You should indicate the efficiency of your primer set.

L.323: If you present a result (survival rate), you should provide a figure.

L.331: The title of Figure 1 should be Final Body Weight and not FBW. The Y axis title should be clearer : Fish body weight (mg). The colours of F25 and F35 are too similar (valid for all graphs).

L.414: You should state in the legend of Figure 8 that NPY stand for Neuropeptide Y.

L.506: Assessment of fish welfare is difficult and should include more than one stress-related gene marker and behavioural analysis (swimming pattern). Therefore, I recommend to not discuss about fish welfare as this require more data and analysis.

L.510: Improved feed intake may also have drawback like myocardial infarction.

L.532-537: Repetition of “As regard”. May be replaced by “considering” or equivalent.

L.550: Is this statement more important for teleost than other animal species ?

L.570: Did you explained what was the signification of ROT group ? Is this rotifer ? You mentioned that all fish from 5 to 10 dpf were fed with rotifer but is ROT group only fed with rotifer through the whole experiment ?

Comments on the Quality of English Language

The English language could be improved. Short sentences will make the manuscript clearer.

Author Response

L.14: Attractants may be replaced by palatable as it seems more appropriate in the sentence context.

We would like to thank the reviewer for the suggestion, but the correct definition to be used in the present study is feed attractant in agreement with the previous paper published on the same topic by Conti et al. 2023.

L.17: Explain how the synthetic flavors are sustainable.

Synthetic flavors can be considered a more sustainable alternative compared to the currently used feed attractants, mainly obtained marine extracts. Due to the unsustainability issues about the use of marine-derived raw materials, modern aquafeeds should be formulated with alternative solutions, like those tested in the present study. In the simple summary, we simply add “more sustainable” due to the concise form of this section. However, this concept has been better described in the Introduction section.

  1. 19: I think attractants and palatable are two distinct things. Attractants aim to stimulate the fish feed intake while palatable ensure that the fish keep and assimilate the aquafeed. Therefore was the study looking more at attractant or palatability of the synthetic flavors ?

Olfaction and gustation detect overlapping sets of attractive molecules (e.g., amino acids). Both chemical attraction and feeding stimulation affect feeding behaviour and have to be taken into account during fish farming. Feeding activity is a consequence of chemical attraction derived by the presence of attractive molecules in the feed which affect both olfaction and gustation. In this sense, since the tested molecules are able to dissolve into the water their first role is on the olfactory system and the second on palatability.

This image can better describe the concept (also see Hancz et al, 2020)

L.20: What was that aim of the repulsive ? Was it like a negative control ?

Yes, the reviewer is right. In reference to the paper Conti et al. 2023, this flavour was selected through a behavioural test using soaked sponges as negative one. However, in the previous paper, this negative effect was not evident when it was mixed with the diets. A similar result was detected in the present study and authors decided to test this flavour initially identified as negative also in the present paper since effects are often related to the fish developmental stage.

In each case the fact that this was a negative control has been added to the MS.

L.23: Fish welfare should be defined.

The definition has been added in the introduction section

L.46: Aquafeed should be define.

Aquafeed” is a widely used term to identify feeds intended for the aquaculture sector. In our opinion, it is not necessary to define it.

L.46-48: This sentence may be rephrased in two sentences to get clearer.

Sentence has been rephrased according to the suggestion.

L.48-51: Same comment. Short sentences are clearer. The ideas are well presented but one at the time will make the manuscript better.

Sentence has been rephrased according to the suggestion.

L.52: A dash between reference 10 and 11 [10-11], will be more consistent and this applies for the whole manuscript.

According to the reference style of the journal, the dash is used to indicate an interval of references (as an example, "8-10" means from 8 to 10). Comma separates two consecutive references (as an example, "10,11")

L.61: “to adapt continuously the response to odorants”.

Corrected

L.73: Is it hedonic and motivational ? Hedonic or motivational ?

This aspect was clarified.

L.125:  In its ingestion, “its” is confusing. If you decide to use “its”, you need also to ass it in front of quality and assimilation to be more consistent.

The reviewer is right, sentence has been corrected.

L.139: The concentration of MS222 is for euthanasia or anaesthesia ? This should be clarified.

The reported concentration was for euthanasia. It has been now clarified in the text.

L.149-155: What was the role of the repulsive (F32-) ?

Please see previous answer about the same topic

L.192-193: How the different organs were conserved until analysis (snap freeze, RNA later) ? Which organs for which type of analysis ?

The information required was already present in each subchapter (ie histology, molecular biology etc), however, in order to make thing clearer a specific sentence has been added at the end of the “Experimental design”.

L.293: If you describe the DNase treatment, you also need to describe how did you inactive the enzyme post treatment.

Since we stated that we used the manufacturer’s protocol to keep the M&M section concise, there is no specific need to describe only the deactivation process.

L.295: How did you assessed that your RNA was not degraded ?

RNA integrity was checked by running 1 µg of total RNA on a 1% agarose gel. A sentence has been added to the manuscript.

L.299-301: You need to state whether you just merged the two housekeeping genes or if you selected the most stable of them.

A sentence has been added. “The mRNA levels of target genes were calculated using the geometric mean of two sta-bly expressed reference genes, using the Bio-Rad CFX Manager 3.1 software. Gene ex-pression changes among groups are reported as relative mRNA abundance.”

L.309-310: You did not mention the cycling parameters for the qCPR reaction, the kit used, the reaction mix, the type of fluorescent (SYBERGREEN ?), the primers concentration…

The reviewer is right. However, since we followed the same procedure described in a previous study, a reference has been added to keep the section short as requested by publishers.

L.311: To guarantee absence of contamination from what ? MiliqWater ? Master Mix ?

Thanks for the suggestion, we clarified in the text that this was referred to the absence of contamination derived from the Master Mix.

L.315: You should indicate the efficiency of your primer set.

The efficiency has been added in the dedicated section.

L.323: If you present a result (survival rate), you should provide a figure.

Results can be shown as text, tables of figures. However, if the reviewer thinks that a figure is more appropriate that can be fine to us.  The graph of the survival rate has been added to the Figure 1.

L.331: The title of Figure 1 should be Final Body Weight and not FBW. The Y axis title should be clearer : Fish body weight (mg). The colours of F25 and F35 are too similar (valid for all graphs).

Figure 1 has been modified according to the reviewer suggestion. The colour of F35 has been changed in all the graphs.

L.414: You should state in the legend of Figure 8 that NPY stand for Neuropeptide Y.

Corrected.

L.506: Assessment of fish welfare is difficult and should include more than one stress-related gene marker and behavioural analysis (swimming pattern). Therefore, I recommend to not discuss about fish welfare as this require more data and analysis.

The reviewer is right, but to assess fish welfare, as stated in the following section we analysed a stress related gene, histopathological indexes and behaviour. This approach has been used and published in many papers about fish welfare.

“In accord to Conti et al. [33], both the solvent (PG) and all the tested synthetic flavors had no adverse effects on the overall fish health and on the target organs usually investigated when novel ingredients/additives are included in fish diets. In fact, no alterations in the intestinal architecture or differences in the histopathological indexes examined were detected; a physiological structure of the hepatic parenchyma was observed in all the experimental groups and no significant differences in the expression of the stress-related marker (nr3c1) was evidenced.”

L.510: Improved feed intake may also have drawback like myocardial infarction.

The authors think that this aspect is out of topic in relation to the paper. In addition, no differences in survival rate were recorded among experimental groups.

L.532-537: Repetition of “As regard”. May be replaced by “considering” or equivalent.

Thank you for the suggestion. Sentence has been accordingly changed.

L.550: Is this statement more important for teleost than other animal species ?

Not necessarily, but we think that the sentence is fine since the paper is about fish.

L.570: Did you explained what was the signification of ROT group ? Is this rotifer ? You mentioned that all fish from 5 to 10 dpf were fed with rotifer but is ROT group only fed with rotifer through the whole experiment ?

The meaning of ROT group was specified in the “Experimental design” section-rotation group (now it has been further clarified). ROT was not for rotifers, which were provided to all the experimental group only from 5 to 10 dpf to ensure a proper larval development.

On this regards we want to underline that this aspect was well defined in the original Ms through the following sentence

“In addition, from 5 to 10 dpf, zebrafish larvae in all the experimental groups were fed the rotifers Brachionus plicatilis (5 individuals/mL) twice a day, according to Zarantoniello et al. [52].” As the reviewer can see no acronym ROT was used for rotifer.

The English language could be improved. Short sentences will make the manuscript clearer.

The whole manuscript has been revised according to the reviewer suggestion.

Round 2

Reviewer 1 Report

Comments and Suggestions for Authors

the article need to revise for reducing the plagiarism. Rest is ok

Comments on the Quality of English Language

a through revision in language is highly recommended 

Author Response

Reviewer 1 – Round 2

The article need to revise for reducing the plagiarism. Rest is ok

Thanks for the suggestions. Materials and methods were the part of the MS with similarity issues. All the sections were checked and eventually simplified by using appropriate references that completely describe the used methods avoiding , at the same time self-plagiarism.

Some parts have not been modified to still satisfy the comments of other reviewers, including English editing.

We hope that the MS is now ready for publication.